

# Radar observability of near-Earth objects using EISCAT 3D

Daniel Kastinen[1,2], Torbjørn Tveito[3], Juha Vierinen[3], and Mikael Granvik[4,5]

[1]Swedish Institute of Space Physics (IRF), Box 812, SE-98128 Kiruna, Sweden
[2]Umeå University, Department of Physics, SE-90187 Umeå, Sweden
[3]UiT - Arctic University of Norway, Postboks 6050 Langnes, 9037 Tromsø, Norway
[4]Division of Space Technology, Luleå University of Technology, Box 848, S-981 28 Kiruna, Sweden
[5]Department of Physics, P.O. Box 64, 00014 University of Helsinki, Finland

**Correspondence:** Daniel Kastinen (daniel.kastinen@irf.se)

**Abstract.** Radar observations can be used to obtain accurate orbital elements for near-Earth objects (NEOs) as a result of the very accurate range and range-rate measureables. These observations allow predicting NEO orbits further into the future, and also provide more information about the properties of the NEO population. This study evaluates the observability of NEOs with the EISCAT 3D high-power large-aperture radar, which is currently under construction. Three different populations
are considered: NEOs passing by the Earth with a size distribution extrapolated from fireball statistics, catalogued NEOs detected with ground-based optical telescopes, and temporarily-captured NEOs, i.e., minimoons. Two types of observation schemes are evaluated: serendipitous discovery of unknown NEOs passing the radar beam, and post-discovery tracking of NEOs using a priori orbital elements. The results indicate that 60–1200 objects per year with diameters $D > 0.01$ m can be discovered. Assuming the current NEO discovery rate, approximately 20 objects per year can be tracked post-discovery near
closest approach. Only a marginally smaller number of tracking opportunities are also possible for the existing EISCAT UHF system. The minimoon study, which used a theoretical population model, orbital propagation, and a model for radar scanning, indicates that approximately 7 objects per year can be discovered using 8–16% of the total radar time. If all minimoons had known orbits, approximately 80–160 objects per year could be tracked using a priori orbital elements. The results of this study indicate that it is feasible to perform routine NEO post-discovery tracking observations using both the existing EISCAT UHF
radar and the upcoming EISCAT 3D radar. Most detectable objects are within 1 LD distance of the radar. Such observations would complement the capabilities of the more powerful planetary radars that typically observe objects further away from Earth. It is also plausible that EISCAT 3D could be used as a novel type of an instrument for NEO discovery, assuming a sufficiently large amount of radar time can be used. This could be achieved, e.g., by time-sharing with ionospheric and space debris observing modes.



# 1   Introduction

All current radar observations of near-Earth objects (NEOs), asteroids and comets with perihelion distance $q < 1.3\,\mathrm{au}$, are conducted post-discovery (Ostro, 1992; Taylor et al., 2019). Radar measurements allow determination of significantly more accurate orbital elements (Ostro, 1994). They may also allow construction of a shape model (Ostro et al., 1988; Kaasalainen and Viikinkoski, 2012) and provide information about composition based on polarimetric radar scattering properties (Zellner and Gradie, 1976). In some cases, the absolute rotation state of the object can also be determined by tracking the scintillation pattern of the radar echoes (Busch et al., 2010).

Radar observations of NEOs are very resource limited, i.e., there are significantly more observing opportunities during close approaches than there is radar time available on the Arecibo and Goldstone DSS-14 radars, the two radars that perform most of the tracking of asteroids on a routine basis. In order to increase the number of radar measurements of NEOs, it is desirable to extend routine NEO observations to smaller radars such as the existing EISCAT radars, or the upcoming EISCAT 3D radar, henceforth abbreviated as E3D, which is to be located in Fenno-Scandinavian Arctic. While these radars may not be capable of observing objects nearly as far away as Arecibo or Goldstone, or performing high-quality range-Doppler images, these radars are able to produce high-quality ranging and perhaps even contribute to discovery of smaller objects.

As a result of the enhanced survey capability with optical telescopes, the discovery rate of NEOs has greatly increased during the last two decades, from 228 NEOs discovered in 1999 to 2436 discovered in 2019. Recent discoveries include significantly more small objects that have close approach distances within 1 LD than discoveries made 20 years ago. It is these objects that are often within the reach of smaller radars. The EISCAT UHF system has in fact already been successfully used to track the asteroid 2012 $DA_{14}$ (Vierinen, 2013), proving the feasibility of these kinds of observations. One of the range-Doppler observations of 2012 $DA_{14}$ using EISCAT UHF is shown in Figure 1. It is expected that NEO observations using E3D will be of similar nature.

When a NEO makes a close approach to Earth it enters a region where Earth's gravity dominates. Most of the time, objects will make one single pass and then leave this region again. In some rare cases, the objects are temporarily captured by the Earth-Moon system (Granvik et al., 2012; Fedorets et al., 2017). These events are called temporarily-captured fly-bys if the object makes less than one revolution around Earth, and temporarily-captured orbiters, or minimoons, if the object makes one or more revolutions around Earth. The existence of a population of transient minimoons in the vicinity of the Earth opens up interesting scientific and technological opportunities such as allowing a detailed characterization of the NEO population in a size range that is otherwise hard to study empirically as well as providing easily accessible targets for space missions (Jedicke et al., 2018; Granvik et al., 2013). However, only two minimoons have been discovered to date, 2006 $RH_{120}$ (Kwiatkowski et al., 2009) and 2020 $CD_3$ (Fedorets et al., in preparation). Hence there is very little observational data about these objects and very basic questions about the population still remain unanswered. Because minimoons, as opposed to generic NEOs, are bound to the Earth-Moon system for a significant period of time, there are more opportunities to track or perhaps even discover them using radar.





The EISCAT radars have already been used for around 20 years to observe the statistical distribution of space debris without prior knowledge of the orbital elements (Markkanen et al., 2009; Vierinen et al., 2019b). Space debris is a collective term for the population of artificially-created objects in space, especially those in orbit around Earth. This population includes old satellites and rocket components and fragments from their disintegration and collisions. There are approximately 16,000 catalogued space debris objects and it is estimated that there are 750,000 objects larger than 1 cm in diameter (Braun, 2018). The capability to detect, track and catalogue these objects is essential to current and future space operations. One of the most useful methods to measure and catalogue space debris is using radar systems. For example beam-park observations, i.e. a single constant pointing direction, with high-power large-aperture radars are an important source of information when modeling the space debris population (Krisko, 2014; Flegel et al., 2009; Banka et al., 2000). The current EISCAT system also has a history of performing significant contributions to space debris observations such as the Chinese anti-satellite event (Markkanen et al., 2009; Li et al., 2012), the Iridium-Cosmos collision (Vierinen et al., 2009) and the Indian anti-satellite event (Vierinen, 2019). The utilitiy of E3D for space debris discovery and tracking has recently been investigated (Vierinen et al., 2017a, 2019a). The study showed that E3D is a capable instrument for observing the space debris object population, due to its phased array antenna system and its multi-static geometry. The focus of this study is to determine if E3D could be similarly used to gain information about the population of NEOs.

The space debris application is very closely related to NEO observations as they both entail discovery and tracking of a population of hard radar targets. Both populations follow a power-law distribution in size, i.e. exponential cumulative growth in number as size decreases. There are, however, several differences. The number-density of NEOs close enough to Earth to be detectable with radar is significantly lower. While observability of NEOs using Arecibo and Goldstone has recently been investigated (Naidu et al., 2016), a similar study has not been done for smaller radars such as E3D. Also, we are not aware of any study of the discovery of NEOs using radar. It is thus desirable to investigate the expected observation capability of this new radar system, so that observation programs that will produce useful data can be prioritized.

E3D is the next-generation international atmosphere- and geospace research radar in Northern Scandinavia. It is currently under construction and is expected to be operational by the end of 2021. E3D will be the first multi-static phased-array incoherent-scatter radar in the world. It will provide essential data to a wide range of scientific areas within geospace science (McCrea et al., 2015). The primary mission for the E3D radar, which has largely defined the radar design, is atmospheric and ionospheric research. However, the radar is also highly capable of observing meteors entering Earth's atmosphere (Pellinen-Wannberg et al., 2016), tracking orbital debris (Vierinen et al., 2019a), and even mapping the Moon (Thompson et al., 2016; Campbell, 2016; Vierinen et al., 2017b; Vierinen and Lehtinen, 2009, e.g.,).

The E3D system will initially consist of three sites — in Skibotn in Norway, near Kiruna in Sweden, and near Karesuvanto in Finland. Each of these sites will consist of about $10^4$ antennas and act as receivers. The Skibotn site will also act as the transmitter with an initial power of 5 MW, later to be upgraded to 10 MW. The E3D design allows for novel measurement techniques such as volumetric imaging, aperture synthesis imaging, multi-static imaging, tracking and adaptive experiments, together with continuous operations. As two of the main features of the system will be high power and flexibility, it is a prime candidate for conducting routine radar observations of NEOs. The technical specifications E3D are given in Sect. 4.

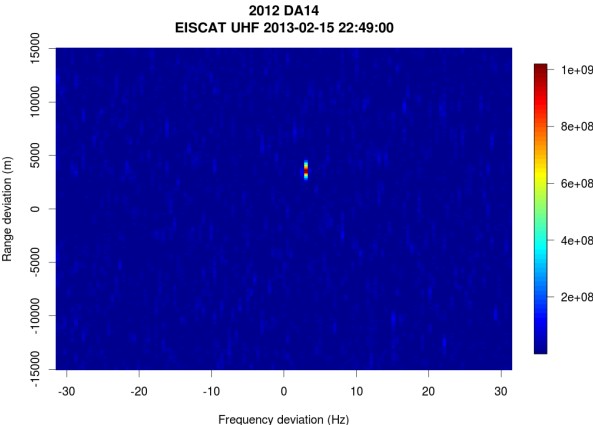

**Figure 1.** Range-Doppler Intensity radar image of asteroid 2012 DA$_{14}$ observed using EISCAT UHF near closest approach on 2013-02-15. The Doppler width and range extent are limited due to the low frequency and limited range-resolution (150 m). No range or Doppler width information apart from object average range and average Doppler is obtained. NEO observations with E3D are expected to be of similar nature, with a factor of 4 times less Doppler resolution due to wavelength, but a factor of 5 more range-resolution due to a higher transmit bandwidth.

In this work, we investigate two possible use cases for E3D for observing NEOs: 1) discovery of NEOs, and 2) post-discovery

tracking of NEOs with known a priori orbital elements. The first case resembles that of space debris observations, where objects randomly crossing the radar beam are detected and, based on their orbital elements, are classified as either orbital space debris or natural objects. The second case is the more conventional radar ranging of NEOs based on a priori information about the orbital elements, which yields a more accurate orbit solution. This is described in detail in Sect. 3.

We estimate the detectability using three different approaches, roughly categorized as first-, second- and third-order. The

first order model uses a power law population density based on fireball statistics (Brown et al., 2002). This model makes some major simplifications about the similarities of the cross-sectional collecting areas of Earth and the E3D radar beam to provide an estimate of the total NEO flux observed by the radar. This method is described in Sect. 5 and the population it uses in Sect. 2.1. In Sect. 6 we describe the second order model that assesses the number of post discovery tracking opportunities that is expected in the near future. For this model, we use close approaches predicted for the last 12 months by the Center for

Near Earth Object Studies (CNEOS) catalogue (JPL, 2020), which is described Sect. 2.2. Lastly, the third order model is a full scale propagation and observation simulation of a synthetic minimoon population which is described in Sect. 7. Although we predict that the vast majority of NEOs are not observable by radars due to size and range issues, one interesting and promising sub-population of NEOs in terms of detectability are the minimoons. This population is described in Sect. 2.3. The results for each method are also given in its respective model description section. Finally we discuss and draw conclusions based on our

results in Sects. 8 and 9.



## 2 Near-Earth-object population models

### 2.1 Fireball statistics

Fireball observation statistics can be used to estimate the influx of small NEOs colliding with Earth. By making the assumption that the flux of NEOs passing nearby Earth is the same, it is possible to make a rough estimate of the number of objects that cross the E3D radar beam. A synthesis of NEO fluxes estimated by various authors is given by Brown et al. (2002), who estimate a log-log linear relationship between the cumulative number $N_{FB}$ of NEOs hitting Earth of diameter $> D$ to be:

$$\log_{10} N_{FB} = a_0 - b_0 \log_{10} D. \tag{1}$$

Here $a_0 = 1.568 \pm 0.03$ and $b_0 = 2.70 \pm 0.08$. For example, using this formula one can estimate that the number of objects colliding with Earth that are larger than 10 cm is approximately $1.8 \times 10^4$ per year.

This population model is convenient, as it will allow us to theoretically investigate the number of objects detectable by a radar, without resorting to large scale simulations.

### 2.2 Known NEOs with close approaches to Earth

The JPL CNEOS maintains a database of NEO close approaches. This database contains objects that have close encounters with Earth, and provides information such as the date and distance for closest approaches (Chesley and Chodas, 2002). As the database consists of known objects, not modelled populations, there are significantly more past encounters than projected future encounters. Many smaller objects, of which only a small fraction are known, are only discovered when they are very close to Earth. As a test population we use the data provided for approaches within 0.05 au that occurred during one year, from 2019-03-13 to 2020-03-13. This population contains 1215 objects, contrasting with 107 objects in the year 2020-03-13 to 2021-03-13. The database was accessed on 2020-03-13 when there were a total of 149,916 objects in the catalogue.

In addition to the closest approach distance, the database provides an estimate of the diameter of each object estimated from the absolute magnitude. The diameter is used to estimate the signal-to-noise ratio (SNR) obtainable for radar observations, both planned and serendipitous discovery observations near the closest approach.

In Fig. 6 we show the distributions of some of the orbital elements in the NEO population being investigated. The reason for the peculiar shape of eccentricity as a function of semi-major axis (top left plot in Fig. 6) is that anything in the bottom-right of the plot, outside the region populated, would not cross Earth's orbit. Similarly, the inclination distribution is biased towards low inclinations, simply due to the fact that these objects are more likely to be near Earth.

The CNEOS catalogue offers a way to judge what a realistic number of tracking opportunities will be for E3D. Because the orbital elements are not known for most smaller NEOs, the primary source for tracking opportunities are newly discovered objects, which are added to the database near closest approach. Approximately 50% of the objects are discovered before closest approach, and 50% after, primarily as the objects are approaching from the direction of the Sun and are not observable in the daylit hemisphere using telescopic surveys. The number of annual detections has been steadily increasing, and we expect





significantly more tracking opportunities within the next few years given the constantly improving sky surveys and the start of new surveys such as the Rubin Observatory Legacy Survey of Space and Time (LSST; Ivezić et al., 2019).

## 2.3 Minimoon model

Only two minimoons have been discovered so far and we therefore have to rely on theoretical predictions of their orbits and sizes rather than a model that is based on direct observational data. The theoretical models are based on a numerical analysis of the NEO capture probability, estimation of the average capture duration, and the estimated flux of NEOs into the capturable volume of phase space. Whereas Granvik et al. (2012) focused on minimoons only and estimated the flux based on the debiased NEO model by Bottke et al. (2002), Fedorets et al. (2017) extended the model to encompass both orbiters and fly-bys and tied

the model to the updated NEO model by Granvik et al. (2016).

    Here we use the newer minimoon model by Fedorets et al. (2017). The average minimoon makes 3–4 revolutions around Earth during a capture which lasts some 9 months. The largest minimoon captured at any given time is about 1 meter in diameter. The realization of the model contains 20,272 synthetic minimoon orbits with absolute V-band magnitude $29.6 < H_V < 37.1$. The epochs of the synthetic minimoons are randomly spread across the 19-year Metonic cycle to properly average

over the changes in the mutual geometry of the Earth, the Moon, and the Sun. Hence about 1000 minimoons are captured in any given year during the simulation. The lead time to capture is typically not more than 2–3 months, so the simulated minimoon environment is in steady state 12 months after the epoch of the chronologically first synthetic minimoon.

    We need to convert absolute magnitudes to diameters to find the radar SNR in subsequent calculations. Using the relationship between the absolute magnitude $H_V$, diameter $D$, and geometric albedo $p_V$, a relation between $H_V$ and $D$ can be derived (e.g.

Harris and Harris, 1997; Fowler and Chillemi, 1992):

$$\log_{10}(D) = 3.1236 - 0.5\log_{10}(p_V) - 0.2H_V. \tag{2}$$

Here it is assumed that the integral bolometric Bond albedo is approximately equal to the Bond albedo in the visual color range, allowing the use of the geometric albedo. Here we assume a geometric albedo $p_V = 0.14$ (Granvik et al., 2012; Morbidelli et al., 2020), which will transform the modeled population in such a way that the distribution is not only shape representative of the

true population but also magnitude representative. In other words, if we simulate $N$ possible object measurements in a certain parameter space region, this corresponds to an expected $N$ real object measurements.

    We also need an estimate of the NEO rotation rate, because it affects the SNR of a radar measurement (Sect. 3). The rotation-rate distribution of objects smaller than 5 meters in diameter is unfortunately not well constrained. The rotation rate appears to increase exponentially with decreasing size (Bolin et al., 2014; Pravec et al., 2002). Virtually unbiased radar observations have

also revealed that very few small NEOs rotate slowly (Taylor et al., 2012; Benner et al., 2015). In what follows we assume that the objects could have one of four different rotation rates: 1 000, 5 000, 10 000 or 86 400 revolutions per day.



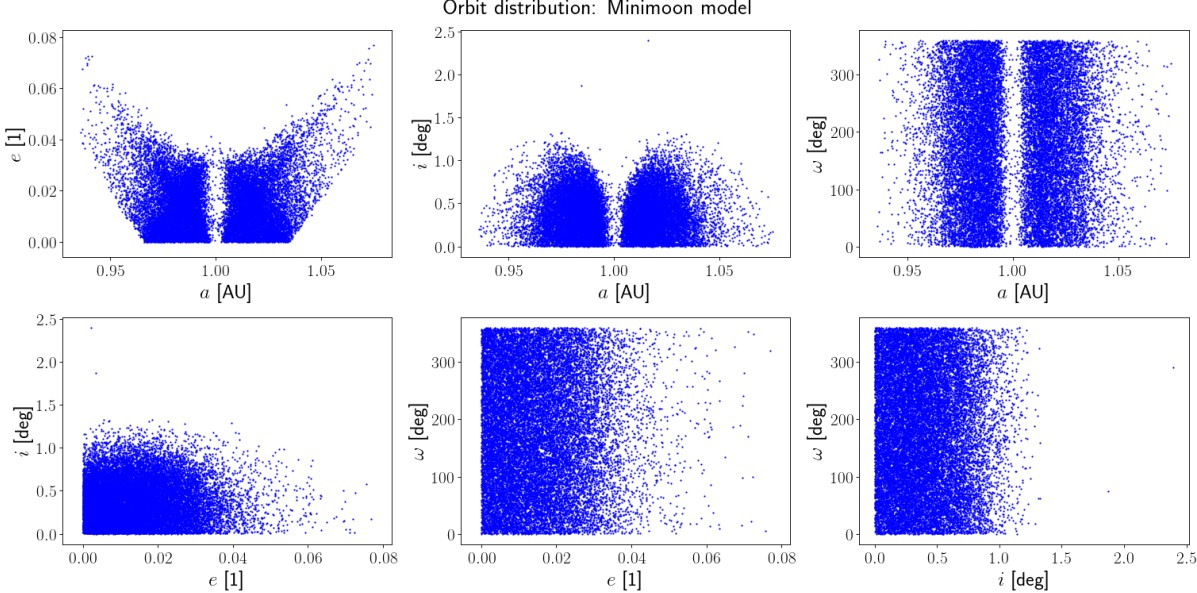

**Figure 2.** Orbital elements of the synthetic minimoons in the heliocentric J2000 frame. This is the initial distribution for the minimoon simulations. We have not illustrated the longitude of the ascending node as this will be closely related to the temporal distribution of objects.

## 3 Radar detectability

When observing NEOs with radar, the most important factor is radar detectability, which depends on the SNR. SNR is determined by the following factors specific to an object: diameter, range, Doppler width, and radar albedo; as well as factors specific to the radar system: antenna size, transmit power, wavelength, and receiver system noise temperature. The following model for radar detectability presented here is similar to the one given by Ostro (1992), with slight modifications and additions.

The Doppler width $B$ of a rotating rigid object depends on the rotation rate and diameter of the object:

$$B = \frac{4\pi D}{\lambda \tau_s}.$$

Here $D$ is the object diameter, $\lambda$ is the radar wavelength, and $\tau_s$ is the rotation period of the object. The Doppler width can be used to determine the effective noise power entering the radar receiver. By coherently integrating the echo $B^{-1}$ seconds, we obtain a spectral resolution that corresponds to the Doppler width. When dealing with a pulsed radar, we also need to factor in the transmit duty-cycle $\gamma$, which effectively increases the noise bandwidth. The noise power can be written as follows:

$$P_N = k T_{\text{sys}} B \gamma^{-1}.$$

Here $k$ is the Boltzmann constant and $T_{\text{sys}}$ is the system noise temperature.





The radar echo power originating from a space object, assuming the same antenna is used to transmit and receive, can be obtained using the radar equation:

$$P_S = \frac{P_{\text{tx}} G^2 \lambda^2 \sigma}{(4\pi)^3 R^4}. \tag{3}$$

Here $P_{\text{tx}}$ is the peak transmit power, $G$ is the antenna directivity, $R$ is the distance between the radar and the object, and $\sigma$ is the radar-cross section of the target.

The radar cross-section of a NEO can be estimated using the radar cross-section of a dielectric sphere, which is either in the Rayleigh or geometric scattering regime (e.g., Balanis (1999)). The transition between the regimes occurs at approximately $0.2\lambda$. This is similar to the approach taken by the NASA size estimation model for space debris radar cross-sections (Liou et al., 2002), which is validated using scale-model objects in a laboratory. Resonant scattering is not included in the model, as the objects are irregular in shape and on average the sharp scattering cross-section resonances are smeared out. As we are dealing with natural objects made out of less conductive materials than man-made objects, the radar cross-section will be scaled down by a factor from that of a perfectly conducting sphere. This factor is a dimensionless quantity called the radar albedo $\hat{\sigma} \approx |(\varepsilon_r - 1)/(\varepsilon_r + 2)|^2$, with $\varepsilon_r$ being the relative permittivity of the object. In this study a commonly used value of $\hat{\sigma} = 0.1$ is used (Ostro, 1992). The radar cross-section model is thus

$$\sigma = \begin{cases} \hat{\sigma} \frac{9\pi^5}{4\lambda^4} D^6 & D < \lambda/(\pi\sqrt{3}), \\ \hat{\sigma} \frac{1}{4} \pi D^2 & D \geq \lambda/(\pi\sqrt{3}). \end{cases} \tag{4}$$

When detecting an object, we are essentially estimating the power of a complex normal random variable measured by the radar receiver. We assume that we know the system noise power to a far better precision than we know the power originating from the space object, and ignore uncertainty in determining the noise power in the error budget. It can be shown that the variance of the power estimate is then

$$\text{Var}\{\hat{P}_S\} = \frac{(P_S + P_N)^2}{K}, \tag{5}$$

where $K$ is the number of independent measurements. It is possible to obtain an independent measurement of power every $B^{-1}$ seconds, which means that there are $K = \tau_{\text{m}} B$ measurements for an observation period of length $\tau_{\text{m}}$, assuming that $\tau_m \gg B^{-1}$.

To determine if the measurement is statistically significant or not, the signal power originating from the standard deviation of the power estimation error relative to the signal power can be used:

$$\epsilon = \frac{P_S + P_N}{P_S \sqrt{K}}. \tag{6}$$

For example, we can use $\epsilon = 0.05$ as the criterion of a statistically significant detection. In this case, the error bars are 5% in standard deviation of the received signal power. We can then determine the number of required samples:

$$K = \frac{(P_S + P_N)^2}{\epsilon^2 P_S^2}. \tag{7}$$





We can also determine the minimum required observation time needed to reduce the relative error to $\epsilon$:

$$\tau_\epsilon = \frac{(P_S + P_N)^2}{\epsilon^2 P_S^2 B}. \tag{8}$$

The commonly used SNR reported for planetary radar targets compares the received power to the standard deviation of the averaged noise floor:

$$\rho = \frac{P_S}{P_N}\sqrt{\tau_m B} \tag{9}$$

In the case of geometric scatter, this is

$$\rho = \frac{1}{4^{\frac{9}{2}}\pi^{\frac{7}{2}}k}\frac{P_{\text{tx}}\gamma G^2\lambda^{\frac{5}{2}}}{T_{sys}}\frac{\hat{\sigma}d^{\frac{3}{2}}\tau_s^{\frac{1}{2}}}{R^4}\tau_m^{\frac{1}{2}}, \tag{10}$$

where the equation is grouped into a constant, a radar dependent term, an object dependent term, and the observation duration.

### 3.1 Serendipitous detectability

The above considerations for detectability of a space object assume that there is a good prior estimate of the orbital elements, which allows radial trajectory corrections to be made when performing the coherent and incoherent averaging. If the objective is to discover an object without prior knowledge of the orbit, one must perform a large scale grid search in the radial component of the trajectory space during detection. In this case, it is significantly harder to incoherently average the object for long periods of time, while matching the radial component of the trajectory with a matched filter. The search space simply would be too large. For space debris targets, we estimate the longest coherent integration feasible at the moment to be about $\tau_c = 0.2$ seconds. This also corresponds to the longest observing interval. We will use this as a benchmark for serendipitous discovery of NEOs. In this case, we need to evaluate the measurement bandwidth using

$$B = \max\left(\frac{4\pi D}{\lambda\tau_s}, \frac{1}{\tau_c}\right), \tag{11}$$

i.e., the bandwidth has a lower bound, which is determined either by rotation rate or by coherent integration length. In most cases, the receiver noise bandwidth will be determined by the coherent integration length $B = \tau_c^{-1}$. We will use this in the subsequent studies of serendipitous detectability.

Assuming that we cannot perform incoherent averaging without a priori knowledge of the orbital elements, the SNR will then be

$$\rho = \frac{P_S}{P_N}, \tag{12}$$

which does not include any effects of incoherent averaging of power.

## 4 EISCAT 3D

The E3D Stage 1 is expected to be commissioned by the end of 2021. It will then consist of one transmit and receive site in Skibotn, Norway (69.340°N, 20.313°E) and two receive sites in Kaiseniemi, Sweden (68.267°N, 19.448°E) and Karesuvanto,





Finland (68.463°N,22.458°E). Each of these sites will consist of a phased array with about $10^4$ antennas, which will allow rapid beam steering.

The transmitter in Skibotn will initially have a peak power of 5 MW, later to be upgraded to 10 MW. For this study, we have
assume a transmit power of 5 MW. The transmit duty-cycle of the radar is $\gamma = 0.25$ or 25%. It will not be possible to transmit continuously with full peak power in a manner that planetary radars conventionally operate. At the same time, the beam on/off switching time for E3D is only a few microseconds, opposed to 5 seconds for DSS-14 and Arecibo (Naidu et al., 2016), which makes it possible to observe nearby objects.

The other key radar performance parameters for the Stage 1 build-up of E3D are: peak radar gain of 43 dB ($G_0$); receiver
noise temperature of 150° K; transmitter bandwidth of $\leq 5$ Hz; transmitter bandwidth of $\leq 30$ Hz; operating frequency of 233 MHz (wavelength of 1.29 m). The main lobe beam width is approximately 0.9°. The system will be able to point down to at least 30° elevation. As the radar uses a planar phased array, the gain reduces as a function of zenith angle $\theta$ approximately as $G = G_0 \cos(\theta)$. The radar will also allow two orthogonal polarizations to be received and transmitted, allowing for polarimetric composition studies of NEO radar cross-sections.

Using Eq. 10, it is possible to compare the sensitivities of different radar systems, as the radar parameter dependent portion is

$$\rho \propto \frac{P_{TX}\gamma G^2 \lambda^{5/2}}{T_{sys}}. \tag{13}$$

Using parameters given by Naidu et al. (2016), the Arecibo observatory 2.38 GHz planetary radar is approximately a factor of $1.7 \times 10^4$ more sensitive than E3D. The Goldstone DSS-14 system, on the other hand, is a factor of 600 more sensitive than
E3D, and E3D, in turn, is approximately a factor of 8 more sensitive than the existing EISCAT UHF.

As E3D will have a lower sensitivity and very short transmit beam on/off switching time compared to conventional planetary radars it may be possible to use it as a search instrument, as it is possible to observe nearby objects and the beam has a large collecting volume. The ability use the phased array antenna to point quickly anywhere with a $120° \times 360°$ Field Of View (FOV) can also be used to increased the effective collecting volume when the radar is used for performing serendipitous discovery of
NEOs.

A full FOV scan can be performed relatively quickly. The beam broadens as the radar points to lower elevations making the scan pattern non-trivial to calculate. At the zenith the beam width of E3D is about 0.9° while at 30° elevation the beam is roughly 4.3° wide. The broadening only occurs in the elevation direction, i.e., the main lobe becomes elliptical instead of circular (Vierinen et al., 2019a). Using the know relation between broadening and elevation, one can estimate that approximately
1250 beam directions are needed to scan the entire FOV of E3D. An example scanning pattern is illustrated in Fig. 3. Using an integration time of 0.2 s the entire sky is scanned every 4–5 minutes.

## 5   Discovery of near-Earth objects

Most $< 1$-m-diameter NEOs pass Earth undetected. In order to estimate the feasibility of discovering these objects using radar, we consider a short coherent integration detection strategy that performs a grid search of radial trajectories similar to space





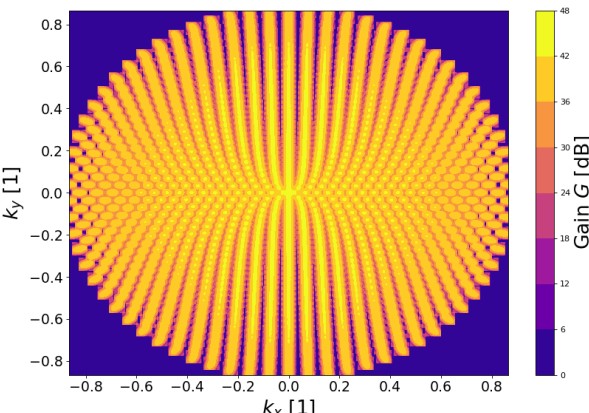

**Figure 3.** An example full FOV radar scan pattern for E3D including the beam broadening effect. The coordinate axis are the normalized wave vector ground projection $k_x, k_y$. This scan pattern use approximately 1250 scan direction of 0.2 seconds each, completing in about 4–5 minutes.

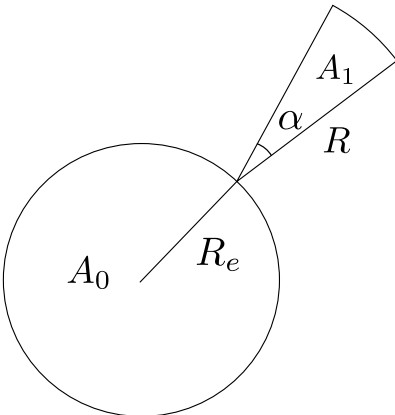

**Figure 4.** Cross-sectional area of NEOs hitting Earth is indicated with $A_0$. The cross-sectional area of NEOs hitting the E3D beam is $A_1$, with the radar antenna beam width $\alpha$ and the maximum detectable range $R$.

debris (Markkanen et al., 2009). In the subsequent analysis, a coherent integration length of 0.2 seconds is assumed, based on the assumption that integration lengths longer than this would not be computationally feasible. The short coherent integration time also makes it possible to ignore the effect of an object's rotation rate on its detectability, as the effective bandwidth of the coherently integrated signal is nearly always larger than the Doppler bandwidth of the object.

    Using the fireball flux reported by Brown et al. (2002), it is possible to estimate the flux of NEOs of various sizes that hit

Earth, as described in Sect. 2.1. We will make the following assumptions: 1) the flux of objects that pass near Earth is the same as the flux of objects hitting Earth, 2) we ignore the effects of Earth's gravity on incoming objects, and 3) assume that all objects approach Earth aligned with the normal to the meridian circle where E3D is located. It is then possible to treat Earth and the





E3D beam as targets with a certain cross-section for incoming NEOs. In this case, Earth is a circular target with a cross-section area $A_0 = \pi R_e^2$ and the E3D radar beam is a target with a cross-section area $A_1 = \frac{1}{2}R(D)\alpha$, where $R(D)$ is the maximum

range at which an object of diameter $D$ can be detected. The maximum detectable range $R(D)$ as a function of diameter is shown in Fig. 5. The beam opening angle is $\alpha$, which assumed to be $1°$. The cross-sectional areas for Earth-impacting NEOs and NEOs passing the radar beam is depicted in Fig. 4.

The cumulative flux of objects larger than diameter $D$ provided in Eq. 1 can be written as

$$N_{FB} = 36.983D^{-2.7}. \tag{14}$$

Differentiating this, we can obtain the density function of objects of diameter $D$:

$$n_{FB} = 99.854D^{-3.7} \tag{15}$$

The flux density of NEOs crossing the radar beam of a certain size or larger can now be roughly estimated as

$$n_{E3D}(D) = \frac{A_1(D)}{A_0}n_{FB}(D), \tag{16}$$

which is in units of objects impacting Earth per year per meter of diameter. When integrated over diameter, we obtain the

cumulative distribution function for the number of radar detections per year of objects with diameter larger than $D$:

$$N_{E3D}(D) = \frac{1}{A_0}\int_D^\infty A_1(D')n_{FB}(D')dD'. \tag{17}$$

The maximum coherent integration time is $\tau_c = 0.2$ seconds. However, it takes significantly longer than that for a NEO to drift across the radar beam. Assuming a transverse velocity across the beam of 40 km/s and detection at a range of $10^4$ km, it takes approximately $\tau = 4.4$ seconds for an object to cross the beam. It is therefore feasible to scan up to $N_b = \tau/\tau_c = 22$

different pointing directions with a fence-like scan to increase the collecting area of the radar. In the optimum case, these directions are independent from one another, thus increasing the cumulative number of detections by a factor of $N_b$. While the beam crossing time is a function of range, we'll use this reprentative value at $10^4$ km.

The cumulative number of radar detections of objects with $D > 1$ cm is estimated to be $\approx 60$ without beam scanning. The lower bound of 1 cm is determined by the minimum detectable object at the height where ablation becomes significant at 100

km. If a fence-like scan with 20 beam pointing directions is used to increase the total effective collecting area of the radar, the number of detections goes up to 1200 detections per year. The cumulative density function of radar detections per year is shown on the right hand side of Fig. 5. The blue line indicates radar detections with a fixed radar beam position, and the orange line indicates cumulative detections per year when using a 20 position fence scan. The dashed black line indicates the transition from geometric scattering to Rayleigh scattering. Objects smaller than the Rayleigh scattering size limit become much harder

to observe due to vanishing radar cross-section, which causes the bend in the cumulative density function.

It is worth noting that the above numbers are very rough estimates based on the above simplistic "shotgun" model. However, the results are very promising, because the magnitude of the number of serendipitous radar detections of NEOs is in the order





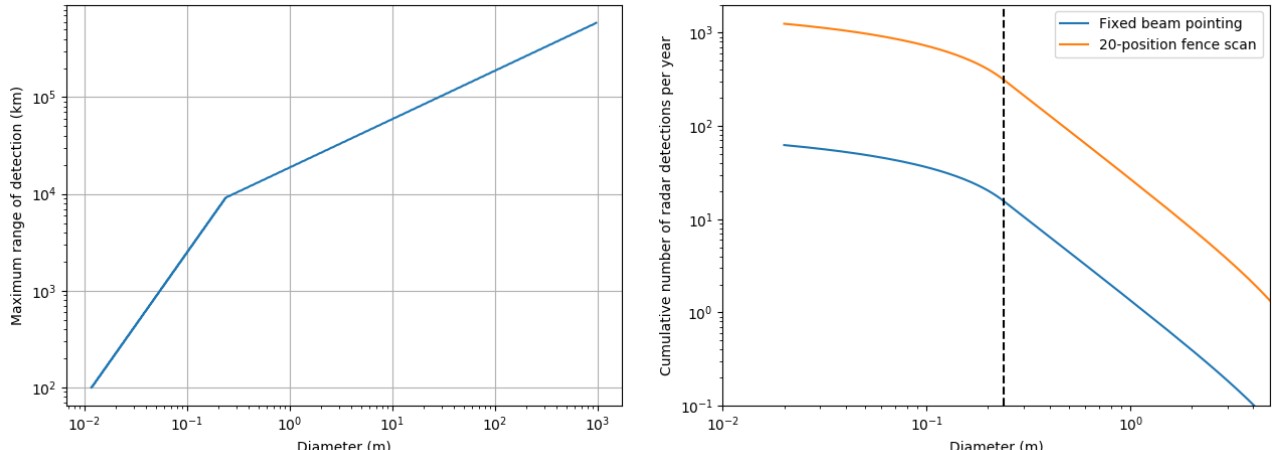

**Figure 5.** Left: Maximum range at which an object of a certain diameter is detectable with E3D with a 0.2 second coherent integration length. The range grows initially faster due to object sizes being in the Rayleigh scattering regime, where the radar cross-section grows proportionally to $\sigma \propto D^6$, where $D$ is object diameter. Once the transition to geometric scattering occurs, the maximum detectable range grows more slowly. Right: The estimated cumulative number of radar detections per year with E3D, assuming 100% time use. The blue line is without beam scanning, and the orange line is with a 20 position fence scan. The black dashed line indicates the transition from Rayleigh to geometric scattering.

of magnitude between 10 and 1000, which is significantly larger than zero. It is therefore plausible that NEOs in the size range $0.01 < D < 1$ m can be detected using the E3D radar. In order to obtain a more accurate estimate of radar detection rates, a

more sophisticated model of the radar needs to be used, together with a realistic NEO population model.

Assuming that objects are in the geometric scattering regime and that the radar antenna aperture is circular, the search collecting area for a radar for a radar is:

$$A_1(D) = \frac{\pi}{32} \sqrt{\frac{\hat{\sigma}}{kB\rho}} \sqrt{\frac{P\gamma}{T}} \eta d_r D \tag{18}$$

Here $B$ is the coherent integration analysis bandwidth, $\rho$ is the minimum SNR required for detection, $d_r$ is the diameter of the

antenna, and $\eta$ is the aperture efficiency. While a larger antenna is still more advantageous, it is not as crucial for this application as it is for tracking or imaging. The number flux density of NEOs that can be detected when crossing the beam $n_{FB}A_1(D)/A_0$ is only linearly dependent on antenna diameter. Factoring everything together, the number flux density of detections per unit diameter is:

$$n_{\mathrm{radar}} = \frac{99.854}{32R_e^2} \sqrt{\frac{\hat{\sigma}}{kB\rho}} \sqrt{\frac{P\gamma}{T}} \eta d_r D^{-2.7} \tag{19}$$





And the cumulative number of radar detections of objects with diameter $> D$ per year, assuming $\hat{\sigma} = 0.1$, $B = 5$, and $\rho = 10$ is:

$$N_{\text{radar}}(D) = 5.261 \times 10^{-4} \sqrt{\frac{P\gamma}{T}} \eta d_r D^{-1.7}, \tag{20}$$

which is valid only when $D > \lambda/(\pi\sqrt{3})$. For example, based on this formula we can estimate the number of NEOs detected by the Arecibo Observatory 430 MHz ionospheric radar. This system has approximately the following radar performance parameters: $P = 10^6$ W, $d_r = 305$ m, $\gamma = 0.05$, $\eta = 0.5$, and $T = 100$ K. It should be possible to detect approximately 45 NEOs per year with a diameter $> 0.15$ m crossing the radar beam, i.e., one detection on average for every 10 days of continuous operations. Many of these objects should be relatively easy to distinguish from satellites and meteors using Doppler shift and range. While this is a low number of expected detections, it may be feasible to search for these objects as a secondary analysis for ionospheric radar observations.

## 6 Observability based on known near-Earth objects

By applying the methods described in Sect. 3 to the population described in Sect. 2.2, we can determine which objects are observable using the E3D radar facility. Both post-discovery tracking and serendipitous discovery was investigated. The observability study was performed in two stages. First, we find the SNR for every object only based on range, size and rotation-rate, without considering whether or not the object is in the radar field of view. We only keep objects with an SNR/hour $> 10$. Then, we use the JPL HORIZONS service to generate an ephemeris for the E3D transmitter location. The ephemeris are used to calculate the maximum SNR when these objects are in the field of view of the radar, keeping only objects with an SNR/hour $> 10$.

In order to estimate SNR, we require the distance between the radar and the object, the object's diameter, and the rotation rate of the object. The CNEOS database contains the minimum and maximum diameter estimates derived from object absolute magnitude. We use the mean of these two diameter estimates. The HORIZONS ephemeris provides distance and elevation angle during times of observation. Rotation rates are not well known, and neither system provides this property for our population of objects.

Bolin et al. (2014) provided two different functions for asteroid rotation period as a function of diameter, based on two different population samples.

$$T_r = 0.005\frac{D}{m} \text{ (hours)} \tag{21}$$

Where $T_r$ denotes the rotation period in hours. This relationship is derived from data of kilometer-size asteroids. Meteor data suggests a much faster rate of rotation:

$$T_r = 0.0001\frac{D}{m} \text{ (hours)} \tag{22}$$



**Table 1.** Objects detectable using E3D during a one year interval.

| Object (ID) | Date (UTC) | Diameter (m) | Distance (LD) | Elevation (deg) | SNR (/0.2s, dB) | SNR (/hour, dB) | Track window (min) | Detection window (min) |
|---|---|---|---|---|---|---|---|---|
| 2019 EN$_2$ | 2019-Mar-14 03:03 | 13.0 | 0.90 | 65.0 | -15.09 | 11.03 | 1945 | - |
| 2020 CD$_3$ | 2019-Apr-04 07:47 | 2.0 | 0.08 | 32.8 | 6.43 | 32.55 | 5195 | - |
| 2019 HE | 2019-Apr-20 23:02 | 19.5 | 0.61 | 37.1 | -8.29 | 17.83 | 2845 | - |
| 2019 JH$_7$ | 2019-May-16 00:11 | 5.2 | 0.17 | 61.2 | 5.38 | 31.50 | 1535 | - |
| 2019 KT | 2019-May-28 03:53 | 20.5 | 0.83 | 66.6 | -9.60 | 16.52 | 1665 | - |
| 2019 LZ$_4$ | 2019-Jun-07 16:55 | 54.0 | 1.36 | 34.9 | -13.81 | 12.31 | 930 | - |
| 2019 LW$_4$ | 2019-Jun-08 15:59 | 15.1 | 0.63 | 80.6 | -6.96 | 19.16 | 2400 | - |
| 2019 MB$_4$ | 2019-Jul-09 04:10 | 27.5 | 0.83 | 42.3 | -9.84 | 16.28 | 2090 | - |
| 2019 OK | 2019-Jul-25 07:52 | 94.0 | 1.49 | 30.1 | -11.78 | 14.34 | 840 | - |
| 2019 OD$_3$ | 2019-Jul-28 03:16 | 18.0 | 0.48 | 56.3 | -2.00 | 24.12 | 1480 | - |
| 2019 ON$_3$ | 2019-Jul-29 02:09 | 11.7 | 0.55 | 49.5 | -9.11 | 17.01 | 335 | - |
| 2019 RQ | 2019-Sep-02 16:50 | 3.2 | 0.28 | 48.9 | -8.12 | 18.01 | 880 | - |
| 2019 SU$_2$ | 2019-Sep-21 03:08 | 4.1 | 0.18 | 67.4 | 3.43 | 29.55 | 795 | - |
| 2019 SS$_2$ | 2019-Sep-21 06:27 | 3.1 | 0.29 | 48.4 | -9.65 | 16.48 | 425 | - |
| 2019 SS$_3$ | 2019-Sep-22 23:48 | 24.5 | 0.73 | 49.6 | -7.46 | 18.66 | 1575 | - |
| 2019 TD | 2019-Sep-29 18:59 | 6.2 | 0.33 | 32.5 | -8.58 | 17.54 | 1200 | - |
| 2019 SM$_8$ | 2019-Oct-01 13:56 | 6.2 | 0.40 | 81.8 | -6.56 | 19.56 | 1085 | - |
| 2019 UG$_{11}$ | 2019-Nov-01 21:02 | 19.0 | 0.54 | 42.4 | -5.33 | 20.79 | 1425 | - |
| 2019 VA | 2019-Nov-02 16:23 | 8.8 | 0.28 | 35.2 | -2.05 | 24.07 | 195 | - |
| 2019 VD | 2019-Nov-04 10:56 | 14.4 | 0.46 | 37.0 | -5.97 | 20.15 | 210 | - |
| 2019 UM$_{12}$ | 2019-Nov-08 17:01 | 49.5 | 1.26 | 50.6 | -10.75 | 15.38 | 1315 | - |
| 2019 VF$_5$ | 2019-Nov-09 23:01 | 13.7 | 0.49 | 33.2 | -8.31 | 17.81 | 155 | - |
| 2019 WG$_2$ | 2019-Nov-23 07:24 | 45.0 | 0.49 | 44.9 | 4.01 | 30.13 | 2420 | - |
| 2019 YB | 2019-Dec-17 23:37 | 5.0 | 0.43 | 60.9 | -10.94 | 15.18 | 1355 | - |
| 2019 YS | 2019-Dec-18 15:27 | 2.1 | 0.16 | 70.5 | 0.20 | 26.32 | 1615 | - |
| 2019 YU$_2$ | 2019-Dec-23 19:53 | 14.4 | 0.25 | 54.7 | 7.15 | 33.27 | 1210 | - |
| 2020 BH$_6$ | 2020-Jan-25 04:51 | 8.6 | 0.17 | 70.7 | 10.72 | 36.84 | 1925 | 50 |
| 2020 CQ$_1$ | 2020-Feb-04 11:46 | 7.3 | 0.16 | 34.4 | 5.81 | 31.93 | 1155 | - |
| 2020 DR$_4$ | 2020-Feb-25 22:10 | 4.9 | 0.26 | 30.8 | -6.84 | 19.29 | 170 | - |

These rotation periods differ by a factor of 50, and as such influence the number of detectable objects a great deal. Assuming

the longer period, we are able to detect 29 out of the 1215 objects. With the shorter estimate, we can only detect 13. We will

be using the longer period estimates from now on.


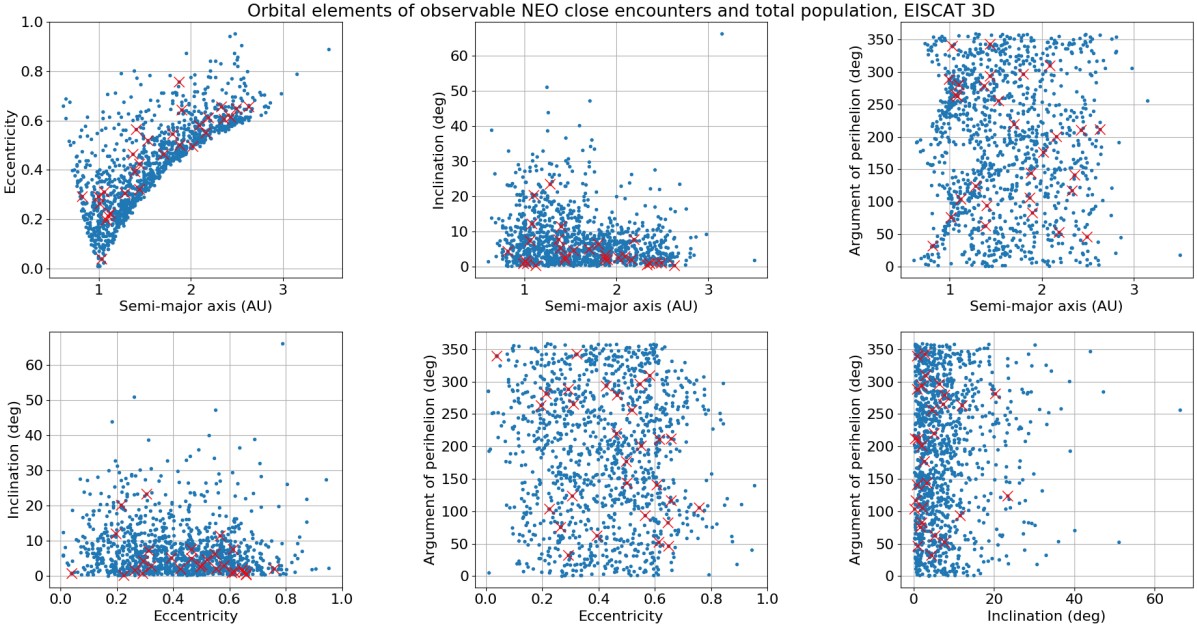

**Figure 6.** Orbital elements of CNEOS database for the period 2019-03-13 to 2020-03-13, with objects having close approaches to Earth and simultaneously observable from the E3D facility marked with red crosses.

A summary of the characteristics of the objects that can be tracked or detected during the studied one year interval is shown in Table 1. Of the the 29 trackable objects 19 were observable in the last half of 2019. The June to December time period featured two thirds of the observable close approach observation windows. This could possibly be explained through the limited view

of the ecliptic plane, as well as small number statistics.

The observable objects were relatively close to the radar, with the shortest range being 0.08 LD and the furthest range being 1.6 LD. The diameters of the observable objects ranged between 2.0 m and 94 m. The highest SNR/hour was 4835, six had an SNR/hour over 1000, and 12 had an SNR/hour over 100. We note that the recently discovered minimoon, 2020 CD$_3$, could have been tracked in April 2019, roughly 10 months before its telescopic discovery.

All observable NEOs were above the 30 degree cut-off elevation for significantly longer than their maximum incoherent integration time estimate. The minimum observation window was 155 minutes, and the maximum was over 5000 minutes. This means that we can expect any serendipitously discoverable objects to be observable for much longer than the time it takes to scan the full field of view, as discussed in Sect. 4. 2020 BH$_6$ would have been discoverable for 50 minutes near its closest approach.

Only a fraction of all objects are discovered and are entered into the CNEOS database. We can assume that there are significantly more objects that could be large enough and have approaches close enough that E3D would discover them with an all sky scan. It should be noted that also 2012 DA$_{14}$ during its 2013 pass could have been easily discoverable with E3D.





Although we have a very limited sample of the total NEO population, it appears that our measurements are not biased towards measuring a specific subset of NEOs with close Earth approaches (Fig. 6). Any potential biases might be revealed
once E3D is in operation, and we can obtain a larger sample space of observable objects.

In order to determine the feasibility of tracking NEO close approaches using the existing EISCAT facilities, we made a similar search for objects observable using the EISCAT UHF radar, which has an antenna gain of $48$ dB, transmit power of $P_{\text{tx}} = 1.8$ MW, system noise of 90 K, and duty-cycle of 12.5%. The total number of observable objects was 17, which was a subset of the objects that could be observed using E3D.

The results indicate that it would be feasible to perform routine NEO post-discovery tracking observations using both the upcoming E3D radar, as well as the existing EISCAT UHF radar. This observing program would nicely complement the capabilities of existing planetary radars, which cannot observe targets that are nearby, due to the long transmit/receiver switching time. Of the $\approx 2000$ NEOs discovered each year, we estimate that approximately 0.5-1%, or approximately 15, can be tracked with E3D or EISCAT UHF, when factoring in that around 50% of the NEOs are discovered before closest approach.

## 7  Observability of minimoons

To accurately determine the observability of a population one needs to construct a chain that considers the following:

1. Model of the measurement system (E3D)

2. Model of the population (minimoons)

3. Temporal propagation of the population (solar-system dynamics)

4. The observation itself (A detection window and SNR)

A recent effort to determine the capability of E3D with regards to space debris measurement and cataloguing produced a simulation software called SORTS (Kastinen et al., 2019). SORTS already includes a model of E3D (item 1) and simulated observations of space debris (item 4) suitable for this study. All the parameters given in Sect. 4 are included in the model of E3D used in the simulation, including realistic antenna gain patterns. SORTS also provides the framework for creating the chain
between the items 1–4 mentioned. Space-debris observations are slightly different from minimoon and NEO observations due to their size, material and orbits. To account for this difference we modified the observation simulation (item 4) according to Sect. 3. Specifically, new SNR formulas from Eqs. 10 and 12 were implemented for use in the NEO and minimoon simulations. The population model previously described in Sect. 2.3 was translated into the format used by SORTS to allow for observability simulation (item 2).

SORTS propagates each object of a given population and searches for time intervals where the object is within the FOV of E3D. We consider the effective FOV of E3D to be a $120°$ cone centered at the zenith, i.e., we allow for pointing down to $30°$ elevation. We do not need to consider any time delays in pointing as interferometric antenna arrays can close to instantaneously point the radar beams electronically.





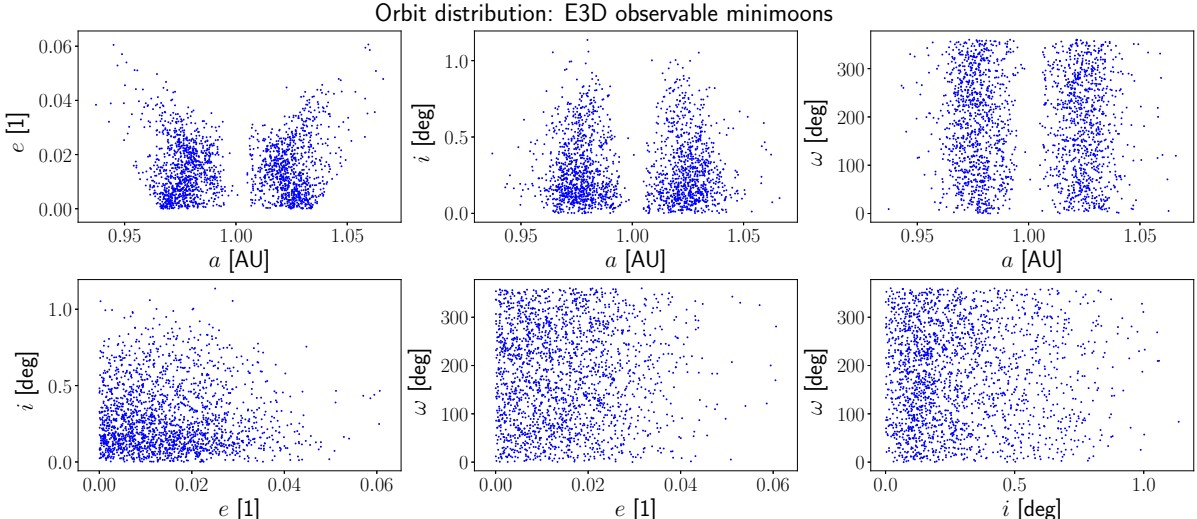

**Figure 7.** Orbital element distribution of minimoons that can be tracked by E3D. This is not the detected orbital element distribution but rather what part of the initial distribution is observable. This illustration should be compared to the initial distribution in Fig. 2.

The only remaining component of the simulation is an interface to a suitable propagation software (item 3). SORTS already
includes propagation software but only for objects in stable Earth orbit, i.e., objects that do not transition to hyperbolic orbits in the Earth-centered inertial frame. We have thus chosen to use the Python implementation of the REBOUND propagator[1] (Rein and Liu, 2012) using the IAS15 integrator (Rein and Spiegel, 2015). This n-body integrator can handle arbitrary configurations of interacting particles and test particles. An interface between the REBOUND propagator and SORTS was implemented allowing for the use of this propagator in all future simulations. For our application we only need to propagate for tens of
years. As such, we have omitted any radiation-related dynamical effects such as radiation pressure and Poynting–Robertson drag as these act on much longer time scales. The integration included all planets and the Moon initialized with the JPL de430 planetary ephemerides.

The integration was configured to use a time step of 60 seconds. This step size allows for decent resolution when searching for viable observation windows by the radar system. The initial state for REBOUND was input in the J2000 heliocentric ecliptic
inertial frame, thus also the output was given in this frame. A standard routine was used to transform to the Earth-centered Earth-fixed J2000 Mean Equator and Equinox frame. In this frame the E3D system is fixed in space and observation windows are readily calculated.



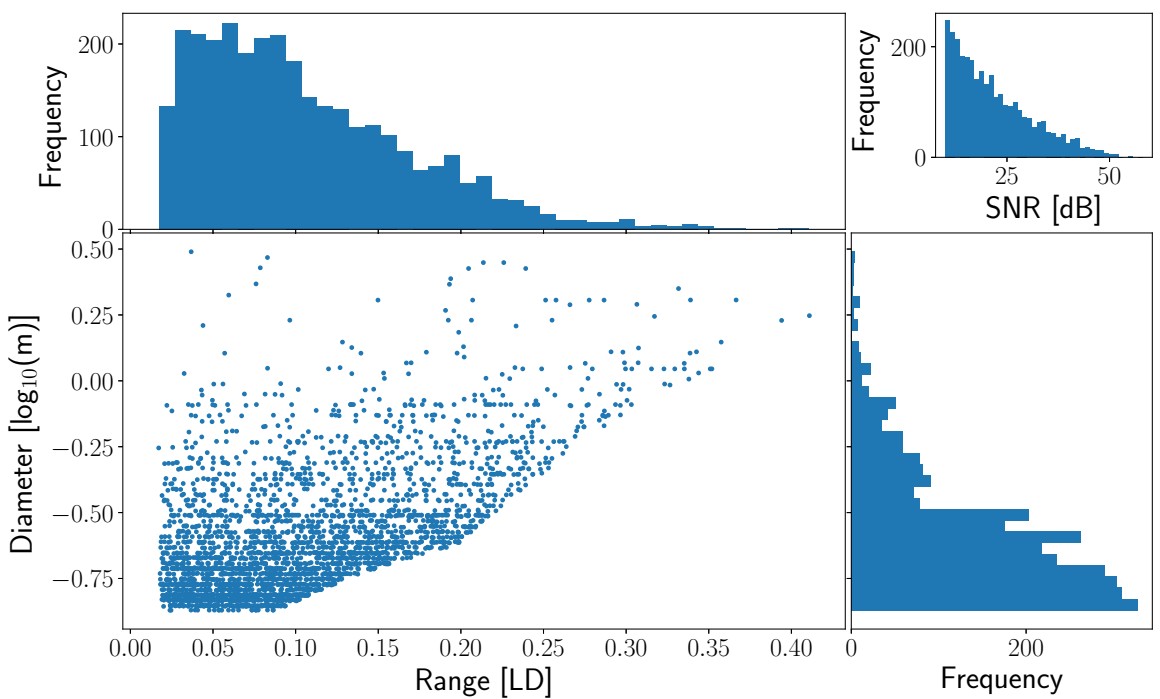

**Figure 8.** Distribution of ranges and sizes of possible observation windows. Included is also the distribution of peak SNR for these observation windows.

## 7.1 Results

All 20,265 synthetic minimoons were integrated for 10 years past their initial epochs. As previously mentioned, we chose to assume that the objects could have one of four different rotation rates: 1 000, 5 000, 10 000 or 86 400 revolutions per day. As such, four different SNRs were calculated for each point in time. It was also assumed that signal integration could not last longer then 1 hour, i.e. $\tau_m$ was set to 1 hour or the observation window length if this was shorter then 1 hour. For each measurement window, only those with at least one measurement point above 10 dB SNR at some rotation rate were saved. A fifth SNR was also calculated based on serendipitous discovery, i.e. with a much shorter coherent integration time of 0.2 seconds. For each of these measurement windows, time series were saved of object position, velocity and all five SNRs.

If we assume that we have a prior orbit for these objects the detections are in essence follow-up tracking measurements and we can consider the tracking SNRs for observability. The a priori orbit does not have to be of good quality, only sufficiently accurate to restrict the search region in the sky. For these tracking measurements, a total of 1 999 out of the 20 265 objects (9.9%) had at least one possible measurement window assuming the 1 000 revolutions per day rotation rate. This number dropped to 7.9%, 7.2% and 5.2% for 5,000, 10,000 or 86,400 revolutions per day respectively.

---

[1] The REBOUND code is freely available at http://github.com/hannorein/rebound

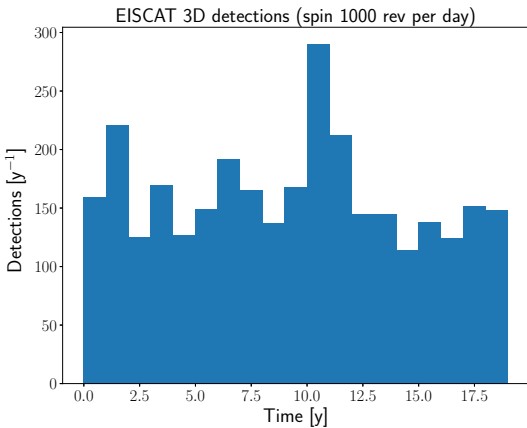

**Figure 9.** Yearly count of possible observation windows of minimoons assuming a rotation rate of 1 000 revolutions per day and that a prior orbit is available.

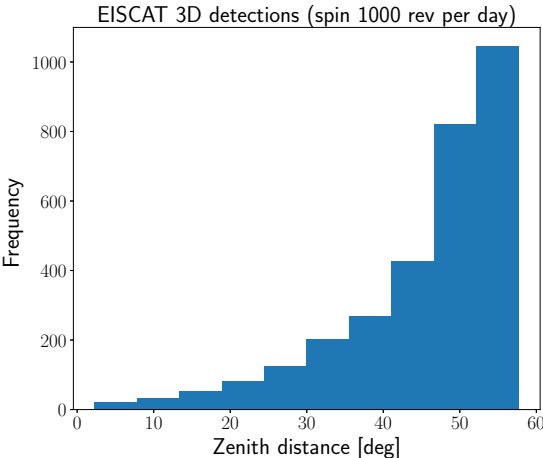

**Figure 10.** The zenith angle distribution of all possible observation windows assuming a rotation rate of 1 000 revolutions per day and that a prior orbit is available.

Without a prior orbit, we have to consider the SNR for serendipitous discovery. Only a total of 116 objects had an observation with 10 dB SNR or more. The rotation rate does not affect this SNR in this case, as the noise bandwidth is determined by the coherent integration time. We assume that the coherent integration time is limited to 0.2 s due to computational feasibility of performing a massive grid search for all possible radial trajectories that matches the trajectory of the target. This results in an effective noise bandwidth of $B = 20$ Hz, when factoring in the 25% duty-cycle. This is nearly always higher than the intrinsic Doppler bandwidth of the target.




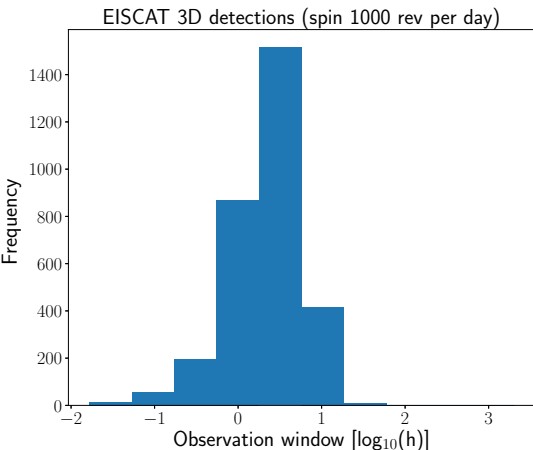

**Figure 11.** The length of each possible observation window that would provide a SNR above 10 dB assuming a rotation rate of 1 000 revolutions per day and that a prior orbit is available.

The distribution of sizes, ranges and SNRs of observable objects are illustrated in Fig. 8. The sharp cut-of in observable minimoon passes is due to the minimum SNR conditions that were imposed. For a single object, according to Eq. 10, the parameter dependant on dynamical integration is range. Thus for an object with a number of possible observation windows, it

is primarily the minimum range of these windows that determine their observability, thereby the lack of observability above a certain range. This equation also contains a transition from Rayleigh to geometric scattering as the diameter increases. This transition is illustrated by the "kink" in the cut-of at approximately 0.24 m in diameter. A promising feature is that even though the majority of detections are made at close range, the lower range of diameters has not been reached by the objects in the model. This indicates the possible existence of smaller objects than those included in the minimoon model that are observable

by E3D. This is supported by the results from Sect. 5 that uses a population model with smaller sizes. There is no apparent dynamical coupling with object size and observation window closest range. This is expected as the orbital dynamics are not a function of size in the simulation or in the minimoon model.

The expected annual detection rate is illustrated in Fig. 9. The distribution is fairly uniform, as expected, and the number of observation windows can thus be averaged over the total time span of the model. Assuming a rotation rate of 1 000 revolutions

per day, the mean expected detection rate is approximately 162 measurement windows per year. The length distribution of these windows with SNR above 10 dB is illustrated in Fig. 11. Generally the observation window is between 1 and 10 hours.

The initial orbital distribution of the observable objects is illustrated in Fig. 7. This illustration should be compared to the initial distribution in Fig. 2. The comparison shows no significant observation biases. However, a dedicated bias study is required for population modeling purposes. It is important to note that the orbits in Fig. 7 are not the detected orbit distribution

as the objects are severely perturbed from Keplerian orbits upon Earth capture.





**Table 2.** Summary counts of the minimoon observability simulations spanning 19 years, and propagated for 10 years. The number of observable objects is representative of the expected total number of real minimoons that can be tracked by E3D in the future if prior orbital elements are known. Each object can have more then one possible observation window and on average each object has approximately 1.5 tracking opportunities. The number of discoverable objects indicates how many serendipitous minimoon detections can be made if an E3D scan hits the object with an integration time of 0.2 seconds. Only a few objects have multiple discovery opportunities.

| Observable objects (out of 20265) | Observation windows | Rotation rate (rev/day) |
|---|---|---|
| 1 999 | 3 081 | 1 000 |
| 1 594 | 2 394 | 5 000 |
| 1 461 | 2 163 | 10 000 |
| 1 058 | 1 529 | 86 400 |
| Discoverable objects | Observation windows | Coherent integration (s) |
| 116 | 128 | 0.2 |

In Fig. 10 we illustrate the zenith distance for the peak SNR observation point of every observation window. As most capturable objects have low-inclination orbits, most observations are centered around the ecliptic, i.e., at low elevation angles for a high-latitude radar.

Summary statistics of the observability study can be found in Table 2. The number of observable objects in Table 2 is
representative of the expected total number of real minimoons that can be tracked by E3D in the future if prior orbital elements are known. Each object can have more than one possible observation window and on average each object has approximately 1.5 tracking opportunities. As the number of discovered NEOs are steadily increasing every year with better instrumentation and analysis techniques, a larger fraction of the population will be discovered, allowing these follow-up tracking measurements to be preformed.

The number of discoverable objects in Table 2 indicates how many serendipitous minimoon detections can be made if a full FOV scan, using an integration time of 0.2 seconds, is continuously performed with E3D. This assumes that the objects passes trough one of the scanning beams at least once. The number of discoverable objects gives an average of 6.74 minimoons discovered per year. As there are only 12 more observation windows than unique objects that are discoverable, a sparse scanning strategy that counts on observing one of many possible passes of the same object is not possible. The distribution of ranges,
sizes, and SNRs of all possible discovery windows are illustrated in Fig. 12.

As discussed in Sect. 4, a full sky scan by E3D could take approximately 5 minutes. As the typical minimoon observation window is on the order of hours, almost all of the possible serendipitous discoveries will be made if the radar performs a full sky scan every hour or half hour. This would use approximately 8–16% of the available radar time.

Conducting routine NEO follow-up observations using E3D would allow for refined orbits and radar measurement's of
hundreds of minimoons every year. Assuming a rotation rate of 1000 revolutions per day and using the observation window

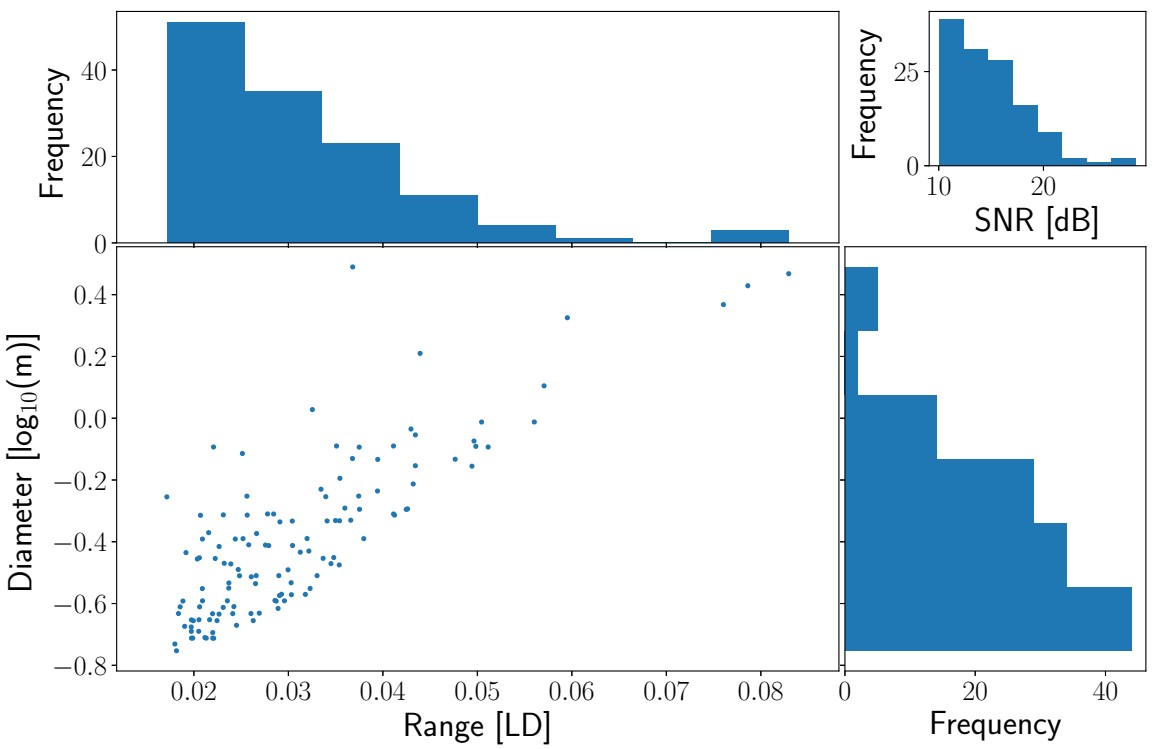

**Figure 12.** Distribution of ranges and sizes of possible discovery windows. Included is also the distribution of peak SNR for these discoveries.

length for each possible tracking window, as illustrated in Fig. 11, we can estimate an average of 502 hours per year would be spent on these observation. This is approximately 5.7% of the total radar time.

## 8 Discussion

For convenience we have summarised the statistics from all three methods that were used to determine the observability of
NEOs with E3D in Table 3. The advantage of using several different methods is not only that one can compare results but also that they inherently peer into different parts of the NEO population.

The fireball observations described in Sect. 2.1 is a sample from the subset of NEOs that make close approaches to Earth. In Brown et al. (2002) arguments are made that the measured population is unbiased. The known NEOs with close approaches described in Sect. 2.2 is also sampling the same subpopulation but in a different size range. This sample is biased and reduced
due to limited observational capabilities. Only a fraction of all NEOs with close approaches are currently detected. If the number of objects as a function of size scale according to Eq. 1, the size ranges listed in Table 3 roughly translate to two orders of magnitude fewer known NEOs than fireballs. As such, the 60–1200 discoveries from fireball statistics versus 1 from the known NEOs with close approaches are mutually consistent.





Based on the CNEOScatalogue, we have found that E3D can be used to observe $\approx 15$ objects per year. This number should
be compared to $\approx 100$ by the Arecibo observatory. Another significant contribution is that smaller-size objects are discoverable
by the E3D system while conducting scans over it's field of view, which opens the potential for a number of discoveries on the
order of 1000 for objects in the $0.01 - 1$ m size range.

The difference in examined populations between the methods suggests that if routine NEO observations are implemented at
E3D, the simulation described in Sect. 7 should be recomputed using a representative subset of the debiased NEO population,
such as the one presented in Granvik et al. (2018) extended to smaller sizes. This would provide guidance for observation
strategy and implementation of analysis, as well as provide observation debiasing for E3D measurements. However, this would
be more costly in terms of computational requirements, as the sampled population would need to be at least 2-3 orders of
magnitude larger than the minimoon population. The implementation of the minimoon observability simulation using SORTS
is fairly general. This means that in the future, the same study can easily be performed for other radar systems as well.

Our results indicate that E3D can provide valuable and unique follow-up measurements of minimoons and NEOs with
close approaches. It also shows that, even though discovering minimoons is sparse, discovery and scanning for the combined
population of minimoons and generic NEOs may be very cost effective as this is inherently dual usage with space debris
observations. I.e., the same radar pulses and survey patterns can be used for discovering objects from all of the above mentioned
populations. Even the discovery of a single new minimoon would be significant since to date only two have been discovered.

The feasibility of a follow-up observation program can be tested in practice by using known space debris objects with large
distances. For example, large objects with Molniya orbits are good candidates for testing the detection capabilities of far-away
objects over long integration times.

It is also valuable to note that E3D will observe over 1000 meteors per hour (Pellinen-Wannberg et al., 2016). Radar meteors
are a direct measurement of the NEO population that makes close approaches to Earth, but at much smaller sizes then the ones
examined here. In Fedorets et al. (2017) 1.46% of all temporarily captured fly-bys and 0.61% of all minimoons impacted Earth.
Due to the tri-static capability of E3D the inferred orbital elements will be of very high quality and it may be possible to trace
meteors back to the minimoon population.

## 9 Conclusions

Our results indicate that it is plausible that E3D can be used to discover NEOs with diameters $D > 0.01$ m. All of the pop-
ulations studied predicted that E3D would discover NEOs by using an all-sky radar survey. A rough estimate of up to 1200
detections per year is possible when using 100% of the radar time at full duty-cycle. This estimate is based on a very simplistic
model, which neglects many important details. However, these results are encouraging and suggest that radar detectability of
NEOs should be investigated further. The capability for discovering NEOs would have several advantages. Radars can observe
in the day-lit hemisphere. The objects that can be found using radar are mostly smaller than the ones detectable using telescopic
surveys and observations of accurate orbital elements of objects in this size range are very scarce.





**Table 3.** Summary statistics from all three methods used to determine the observability of NEOs with E3D.

| Method → | Fireball statistics | NEO close encounter database | | Minimoon simulations | | | |
|---|---|---|---|---|---|---|---|
| Rotation model → | None | $T_r = 0.005 \frac{D}{m} h$ | $T_r = 0.0001 \frac{D}{m} h$ | $1{,}000 \frac{rev}{d}$ | $5{,}000 \frac{rev}{d}$ | $10{,}000 \frac{rev}{d}$ | $86{,}400 \frac{rev}{d}$ |
| Input population size | Density function | 1,215 | | 20,272 | | | |
| Size range | 0.02 m – 100 m | 1.4m – 715 m | | 13 cm – 4.3 m | | | |
| Detectable objects | - | 29 | 13 | 1,999 | 1,594 | 1,461 | 1,058 |
| Detection rate $\frac{1}{y}$ | - | 29 | 13 | 162 | 126 | 114 | 80 |
| Detection E3D usage | - | 0.3%–8% | - | 5.7% | 4.9% | 4.5% | 3.1% |
| Discoverable objects | - | 1 | | 116 | | | |
| Discovery rate $\frac{1}{y}$ | 60–1200 | 1 | | 6.7 | | | |
| Discovery E3D usage | 100% | 8–16% | | 8–16% | | | |

The study of the minimoon subset of the NEO population indicates that a significant fraction of objects could be tracked, with 80–160 observing opportunities per year, assuming that the objects have been previously discovered. There is currently only one minimoon in Earth orbit, but it is no longer observable using EISCAT UHF or E3D due to the long range when the object is in the radar field of view. However, there will be more opportunities in the future for such observations as new minimoons are discovered (Fedorets et al., 2020). Our study shows that an E3D based radar search for minimoons is one potential way for discovering these objects. Our simulation suggests that there are approximately 7 discoveries per year with a 8–16% utilization of radar resources. In addition to utilizing dedicated observing modes for these searches, it should be feasible to also perform a secondary analysis to search for NEOs when running the radar in ionospheric mode.

Our study shows that establishing a post-discovery NEO tracking program that uses close-approach predictions is feasible. Such an initiative could already be commenced with the existing EISCAT UHF radar, which is only slightly less sensitive than the upcoming E3D radar for this purpose. We estimate that roughly 0.5% to 1% of the 2000 objects discovered annually could be tracked using the EISCAT UHF or E3D radars, based on close approaches in 2019. The need for radar resources is minimal, only a few 4–8 hour observing windows each month. However, the observations would need to be scheduled on a short notice, using an automated alert system that notifies of upcoming observing possibilities (cf. Solin and Granvik, 2018). The measurements would yield accurate orbital elements of NEOs, but possibly also estimates for sizes and rotation rates. The dual polarization capability of E3D could also be used to study the composition of these objects.

*Author contributions.* Daniel Kastinen performed the numerical simulations of minimoon observations by E3D in Sect. 7. Torbjørn Tveito performed the detectablility calculation for objects within the CNEOS catalogue in Sect. 6. Juha Vierinen provided the SNR calculations in Sect. 3 and estimated the number of NEOs serendipitously detectable in Sect. 5. Mikael Granvik provided the minimoon population model described in Sect. 2.3. All the authors contributed to the preparation of the manuscript and interpretation of the results.





*Competing interests.* Juha Vierinen is on the editorial board of the journal.





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
