# Peer review of "Radar observability of near-Earth objects using EISCAT 3D"

_Annales Geophysicae, 2020_

## Referee Comment (RC1) · Anonymous Referee #1 · 6 May 2020

The manuscript investigates the expected detectability of near-Earth objects using EIS-CAT 3D. The manuscript specifically studies the detectability of "new" NEOs, extrapolated from the fireball population, known NEOS that come within a lunar distance, and temporarily captured NEOs. The manuscript adequately discusses the methods to extrapolate the expected number of NEOs to be observed by EISCAT 3D. As such, the manuscript is of value to motivate studies that are currently outside of the primary object of EISCAT 3D. Further, the manuscript makes the case that EISCAT 3D will be able to not only detect, but also discovery NEOs. I suggest publication and provide some minor technical corrections below.

Line 173: Equations 2 and 3 are currently not numbered. Line 171: Note that equation 1 is missing the cos(theta) factor for the pole location, such that if you observe

[Figure]

the object parallel to the pole, the bandwidth is near 0, but if you observe the object perpendicular to the pole the bandwidth is maximized. Line 205: For equation 6, is epsilon an alpha level / test statistic? I would caution using epsilon here as the variable since on line 192 it is also used for permittivity, as is typical in radar. Line 245: "transmitter bandwidth of $\leq$ 5 Hz; transmitter bandwidth of $\leq$ 30 Hz", whats the difference here? Line 246: This is the first mention of the operating frequency of the radar. I would suggest to mention this much earlier, potentially in the abstract. It would also be valuable in the introduction to compare this with the operating frequencies of Arecibo and Goldstone. Line 312: "for a radar for a radar" - repeat Line 332: "discovery was investigated" - should be "were" investigated

---

## Referee Comment (RC2) · Peter Brown (Referee) · 2 Jun 2020

This paper provides theoretical estimates of the detectability of small NEOs by radar, specifically small NEO flybys and minimoons. Moreover, it examines both the feasibility and expected detection rate for NEOs using radar for search as well as the ability of EISCAT3D in particular to follow-up small NEOs detected from the ground. While the paper focuses on E3D as a NEO detection instrument it also explores and presents generic algorithms for radar detection of NEOs, with an emphasis on small NEOs. I very much enjoyed reading the paper; it is well written and the first to my knowl-

edge to quantitatively explore radar as a search tool for small NEOs /minimoons. It is clearly a valuable contribution to the field. In particular, the use of both a full population model of NEOs/minimoons extrapolated down to decimeter sizes coupled to a E3D detection model and various considerations/choices for detection modes/integration windows through simulation software is a particularly valuable approach for optimizing both E3D and other radars for NEO/minimoon detection. I recommend publication in essentially the current form.

A few minor technical items/questions the authors may wish to address:

For the case of mini-moon detection, no discussion is given of how to distinguish natural objects (which are comparatively rare) from much more common artificial objects in geocentric orbit. Can the authors comment on the challenges this would pose for radar detection of mini-moons?

At the smallest sizes (sub-meter to decimeter) there is the question of what is the scientific value of radar detections? For such small objects can we estimate rotation rates or surface roughness or is the SNR too low? What can we learn from detecting such small objects which we would not learn but studying them ablate in the atmosphere?

The introduction is very complete, but the authors may want to consider adding or discussing the one reference to the only published paper relating to attempted radar detection of decimeter sized NEOs/meteoroids: Kessler D. J., Landry P. M., Gabbard J. R., and Moran J. L. T. 1980. Ground radar detection of meteoroids in space. In In: Solid particles in the solar system; Proceedings of the Symposium, edited by Halliday I., and McIntosh B. A. IAU. p. 137.

Rotation rates of decimeter-sized meteoroids have been discussed and modelled in a few works such as: Čapek D. 2014. Rotation of cometary meteoroids. Astronomy and Astrophysics 568:1–8. Beech M., and Brown P. G. 2000. Fireball flickering: the case for indirect measurement of meteoroid rotation rates. Planetary and Space Science 48:925–932.

Line 245: transmitter bandwidth of < 5 Hz; transmitter bandwidth of < 30 Hz – I think the latter should be receiver bandwidth?

---

## Author Comment (AC1) · 11 Jun 2020

Dear Reviewer #1, Thank you very much for the helpful feedback on the manuscript. We have below addressed each of the points mentioned in the review individually.

**Line 173: Equations 2 and 3 are currently not numbered.** Fixed.

**Line 171: Note that equation 1 is missing the cos(theta) factor for the pole location, such that if you observe the object parallel to the pole, the bandwidth is near 0, but if you observe the object perpendicular to the pole the bandwidth is maximized.** The formula in equation 1 is actually an approximate formulation of the of the measured Doppler bandwidth where the rotation of the observing frame

(i.e. the rotation of the Earth) is removed as this factor is much smaller then the effect of the intrinsic rotation of the object around its own axis. The text before equation 1 states "depends on the rotation rate and diameter of the object", i.e. not the rotation of the observer. This is slightly unclear and we have added a clarification and changed the preceding text to the following:

*The measured Doppler bandwidth is a combination of relative translation and rotation of the observing frame and the intrinsic rotation of the observed object around its own axis. However, in all cases considered by this study, the effect of a moving observation frame is negligible. As such, the Doppler width $B$ of a rotating rigid object depends on the rotation rate around its own axis and diameter of the object:*

**Line 205: For equation 6, is epsilon an alpha level / test statistic? I would caution using epsilon here as the variable since on line 192 it is also used for permittivity, as is typical in radar.** $\epsilon$ is the relative standard error of the signal power estimator. As such it is not a statistical test but rather an acceptance criteria on the relative standard error. As the $\epsilon$ variable is only used in this section we have changed it to $\delta$ for clarity. We have not used $\alpha$ as this is not hypothesis testing or confidence interval calculations. To clarify this we have added the following to the text before this equation:

*To determine if the measurement is statistically significant or not, a criterion can be set on the relative standard error $\delta$. Using the signal power estimator variance from Eq. 6, $\delta$ is defined as*

**Line 245: "transmitter bandwidth of $\leq$ 5 Hz; transmitter bandwidth of $\leq$ 30 Hz", whats the difference here?** This was a typo. There was also a missing MHz and It should read: "transmitter bandwidth of $\leq 5$ MHz; receiver bandwidth of $\leq 30$ MHz;". It has been changed to this.

**Line 246: This is the first mention of the operating frequency of the radar.**

**I would suggest to mention this much earlier, potentially in the abstract. It would also be valuable in the introduction to compare this with the operating frequencies of Arecibo and Goldstone.**
We have added the operating frequency and peak power of EISCAT 3D in the abstract. We have also added the the operating frequencies and peak power of the EISCAT 3D, Arecibo and Goldstone systems where they are first mentioned in the introduction for the convenience of the reader. While we do in several places state that Arecibo and Goldstone are significantly more sensitive radars than EISCAT 3D, we have chosen not to go into too much detail. The primary purpose of this paper is to study the capabilities of EISCAT 3D. The performance of Goldstone and Arecibo are relatively well characterized e.g., in the paper by Naidu et.al., (2016), which we refer to. We do agree that it would be advantageous to make a more thorough comparison of the performance of various ground based radars for NEO detection in the future, but we are of the opinion that this is outside the scope of this current paper.

**Line 312: "for a radar for a radar" - repeat** Fixed.

**Line 332: "discovery was investigated" - should be "were" investigated** Fixed.

---

## Author Comment (AC2) · 11 Jun 2020

Dear Peter, Thank you very much for the great feedback on the manuscript! We have treated the individual items below.

**For the case of mini-moon detection, no discussion is given of how to distinguish natural objects (which are comparatively rare) from much more common artificial objects in geocentric orbit. Can the authors comment on the challenges this would pose for radar detection of mini-moons?**

Generally space-debris are in a set of well defined orbital regions close to the Earth, orbits that are wildly different from those of minimoons and most NEOs. Therefore the range will be the primary identification as there is very little debris outside 3,000 km

altitude, with the exception of GEO and GTO. As most mini-moon detections will occur much further away these are very likely not artificial object. However, the most secure method of identification is trough the orbit of the detected object. It should be relatively straight forward to distinguish them from artificial objects if the orbit determination is of sufficient quality, especially if the radar astrometry allows for the estimation of the area-to-mass ratio which is typically used to discern between natural and artifical bodies (Jedicke et al. 2018). For the case of EISCAT 3D, describing how to do orbit determination of sufficient quality after initial discovery is a complete separate study in itself but it is definitely possible.

We have added the following on this to the discussion:
*It was shown in Kastinen et al. (2019) that E3D is expected to regularly detect hard target echoes from space debris and other artificial objects in Earth orbit. NEOs and minimoons need to be robustly separated from these artificial objects for discovery operations to be successful. Space debris is mainly confined to two regions: close to Earth ($<3 \times 10^3$ km altitude) or close to geostationary orbit ($\sim 3.6 \times 10^4$ km altitude) (Krisko, 2010; Flegel et al., 2009). Our results show that the typical minimoon or NEO detection will be made at altitudes larger than these regions and up to $3.8 \times 10^5$ altitude. Thus, range to the target can be used as an initial NEO and minimoon identification. If the orbit of the object can be determined, this would be a very reliable method of identification as NEOs and minimoons generally have vastly different orbits compared to space debris (Fedorets et al., 2017).*

**At the smallest sizes (sub-meter to decimeter) there is the question of what is the scientific value of radar detections? For such small objects can we estimate rotation rates or surface roughness or is the SNR too low? What can we learn from detecting such small objects which we would not learn but studying them ablate in the atmosphere?**

First off, it is important to note that the populations that are mainly discussed here will not ablate in the atmosphere but rather pass the Earth by. E.g., only a very small fraction of the intermediate NEO population used to simulate the minimoon population will ablate in the atmosphere, as was shown in [Fedorets et. al 2017 Orbit and size distributions for asteroids temporarily captured by the Earth-Moon system]. With that in mind, the main contribution is that this is a population otherwise not observable! For the case of EISCAT 3D, it is not really the SNR that limits if a measurement of rotation rate or surface roughness can be made but rather the wavelength of the radar relative the target size. Therefore its not expected that it can derive such parameters for smaller objects. Rather, what phased array radars can do really well is possible mono-static orbit determination, good collection area by scanning, low restriction on observing conditions (clouds and sunlight do not impact measurements), the detection of small sizes and easier observational bias calculations. I.e. they produce "a lot" (this is relative of course) of high quality data that can be used for good population density estimates.

We have added the following to the discussion with respect to the above:
*The scientific gain from tracking operations at E3D can be summarized as: efficient, high quality orbit determination and, if the target is sufficiently larger than the wavelength, novel data on surface properties and rotation rates. There are currently not many methods that can discover smaller NEOs unless they collide with the Earths atmosphere, as shown by the low number of discovered minimoons. As such, the scientific gain from discovery operations is in essence the discovery itself, i.e. observation capability of a population otherwise not observable. If the objects are larger, they can be detected with higher probability using optical methods. In these cases radar observations are still valuable for the same reasons as tracking operations are.*

**The introduction is very complete, but the authors may want to consider**

**adding or discussing the one reference to the only published paper relating to attempted radar detection of decimeter sized NEOs/meteoroids: Kessler D. J., Landry P. M., Gabbard J. R., and Moran J. L. T. 1980. Ground radar detection of meteoroids in space. In In: Solid particles in the solar system; Proceedings of the Symposium, edited by Halliday I., and McIntosh B. A. IAU. p. 137.**
The following has been added to the introduction:

*Smaller radars can be used for nearly continuous observing, and it is possible that they can even contribute to the discovery of NEOs. Kessler et al. (1980) presented an early attempt at discovering meteoroids outside of the Earths atmosphere using a space surveillance radar. However, the observation span was only 8 hours and the results were inconclusive, but 31 objects were identified as possible meteoroids. No followup studies were conducted.*

**Rotation rates of decimeter-sized meteoroids have been discussed and modelled in a few works such as: Capek D. 2014. Rotation of cometary meteoroids. Astronomy and ËĞ Astrophysics 568:1–8. Beech M., and Brown P. G. 2000. Fireball flickering: the case for indirect measurement of meteoroid rotation rates. Planetary and Space Science 48:925–932.**
These studies are very interesting! We have added the following to the discussion on rotation rates in the 2.3 Minimoon model section:

*In what follows we assume that the objects could have one of four different rotation rates: 1 000, 5 000, 10 000 or 86 400 revolutions per day. These values are also consistent with modeling of cometary meteoroids. For example, Capek (2014) studied the distribution of rotation rates of meteoroids ejected from 2P/Encke and found that objects with diameters between 1 to 10 centimeters have rotation rates approximately between 10 Hz to 0.1 Hz. There are also indirect observations of meteoroid rotation rates derived from optical meteor light-curve oscillations. Beech and Brown (2000) estimateâĹij20 Hz and less for objects larger than 10 centimeters in diameter.*

**Line 245: transmitter bandwidth of < 5 Hz; transmitter bandwidth of < 30 Hz – I think the latter should be receiver bandwidth?**
Yes, that is right! There was also a missing MHz and It should read: "transmitter bandwidth of $\leq 5$ MHz; receiver bandwidth of $\leq 30$ MHz;". It has been changed to this.